# Clinical Markers of Chronic Hypoxemia in Respiratory Patients Residing at Moderate Altitude

**DOI:** 10.3390/life11050428

**Published:** 2021-05-10

**Authors:** Rosario Fernández-Plata, Ireri Thirion-Romero, Karol J. Nava-Quiroz, Gloria Pérez-Rubio, Sebastián Rodríguez-Llamazares, Midori Pérez-Kawabe, Yadira Rodríguez-Reyes, Selene Guerrero-Zuñiga, Arturo Orea-Tejeda, Ramcés Falfán-Valencia, Rogelio Pérez-Padilla

**Affiliations:** 1Departamento de Epidemiología y Estadística, Instituto Nacional de Enfermedades Respiratorias Ismael Cosio Villegas, Mexico City 14080, Mexico; rosario.fernandez@iner.gob.mx; 2Departamento de Investigación en Tabaquismo y EPOC, Instituto Nacional de Enfermedades Respiratorias Ismael Cosio Villegas, Mexico City 14080, Mexico; draisadora.thirion@gmail.com (I.T.-R.); sebastianrolla@gmail.com (S.R.-L.); midori-pk@hotmail.com (M.P.-K.); 3HLA Laboratory, Instituto Nacional de Enfermedades Respiratorias Ismael Cosio Villegas, Mexico City 14080, Mexico; knava@iner.gob.mx (K.J.N.-Q.); gperezrubio@iner.gob.mx (G.P.-R.); 4Sleep Clinic, Instituto Nacional de Enfermedades Respiratorias Ismael Cosio Villegas, Mexico City 14080, Mexico; ygrr@yahoo.com.mx (Y.R.-R.); seleguerrero@hotmail.com (S.G.-Z.); 5Departamento de Cardiología, Instituto Nacional de Enfermedades Respiratorias Ismael Cosio Villegas, Mexico City 14080, Mexico; aorea@iner.gob.mx

**Keywords:** hypoxemia, HIF-1α, EPO, VEGF, supplemental oxygen, hypoxia

## Abstract

Supplemental oxygen (SO) increases survival in hypoxemic patients. In hypoxia, mammals respond by modulating O_2_-sensitive transducers that stabilize the transcription factor hypoxia-inducible factor-1-alpha (HIF-1α), which transactivates the genes that govern angiogenesis and metabolic pathways. Residing at high altitudes exposes millions of people to hypoxemia with potential adverse consequences on their health. We aimed to identify markers of hypoxemia that can be used in the evaluation of patients in addition to pulse oximetry and arterial blood gases, especially those that could respond after 1 month of oxygen use. We performed a prospective pilot study at 2240 m above sea level, with repeated measurements before and after (b/a) 1-month home oxygen therapy in 70 patients with lung diseases, of which 24/20 have COPD, 41/39 obstructive sleep apnea (OSA), and 5/2 with interstitial lung diseases (ILD), all of them having chronic hypoxemia, as well as 70 healthy subjects as controls. Proteins evaluated included HIF-1α, vascular endothelial growth factor (VEGF), and erythropoietin (EPO). Among the main results, we found that hypoxemic patients had normal levels of HIF-1α but increased EPO compared with healthy controls. VEGF levels were heterogeneous in the sample studied, similar to the control group in COPD, slightly increased in OSA, and decreased in fibrosis. With oxygen treatment, the HIF-1α and EPO decreased in COPD and OSA but not in fibrosis, and VEGF remained constant over time. In conclusion, erythropoietin and HIF-1α identified hypoxemia initially and responded to oxygen. In pulmonary fibrosis, HIF-1α, EPO, and VEGF increased with oxygen therapy, which is likely linked to the disease’s pathogenesis and clinical course rather than hypoxemia.

## 1. Introduction

Chronic or intermittent hypoxemia reduces lung disease subjects’ survival and worsens their health [1,2]. Chronic hypoxemia has a profound effect on patients’ functional capacity and physical activity with COPD and other chronic diseases, not just respiratory [3,4].

On the other hand, supplemental oxygen (SO) treatment improves survival and quality of life in patients with chronic obstructive pulmonary disease (COPD) [1,2], with such relevant results that the criteria used for oxygen therapy in COPD trials in cities near sea level, based on a PaO_2_ < 55 mm Hg, or <60 with polycythemia or pulmonary hypertension, have been applied to patients with other chronic lung diseases and at different ages and altitudes. Hypoxemia generates broad responses of the body that impact cells, tissues, organs, and physiological systems triggered by hypoxia-induced factor-1 alpha (HIF-1α) [5,6] after activating many genes that subsequently determine known physiological responses to hypoxemia, including pulmonary hypertension or polycythemia [3]. The therapeutic interventions are only restoring PaO_2_ by descending to low altitudes or by oxygen therapy [1,2,3]. In chronic intermittent hypoxemia (CIH), such as occurs in sleep apnea, it differs from continuous hypoxia in the activation of HIF-1α. It suppresses HIF-2-mediated transcription and thus peripheral mechanisms, such as an increase in the generation of reactive oxygen species in the chemosensory reflex, which is central in the development of systemic arterial hypertension, which is one of the most frequent chronic diseases [7,8,9].

Chronic hypoxemia is common in residents at moderate altitudes, such as that in the Valley of Mexico at 2240 m above sea level. In a population-based study, 6% of participants aged 40 or over residing in Mexico City found a SpO_2_ of 88% or less and would have in principle indication of oxygen therapy, although only 8% of them received it [10].

Our study’s primary objective is to identify possible biomarkers of hypoxemia that could be used clinically, especially those capable of identifying the impact of oxygen use for 1 month.

## 2. Materials and Methods

The project was approved (approbation code number: C56-16) by the Institutional Ethics Committee of the Instituto Nacional de Enfermedades Respiratorias (INER) in Mexico City, where it was carried out. All participants were adults and signed the corresponding informed consent. The INER is a public healthcare and research institution that receives primarily uninsured patients with respiratory problems.

We planned a prospective, longitudinal study comparing before and after improving patient oxygenation using home oxygen for 1 month. Patients with sleep apnea were treated with nasal continuous positive pressure (nasal CPAP with or without supplemental oxygen).

For this study, there was a selection of 70 patients with hypoxemia (PaO_2_ ≤ 55 mm Hg, or PaO_2_ < 60 with polycythemia or pulmonary arterial hypertension) with repeated measurements before and after (b/a) 1-month home oxygen therapy [1,2] of variable severity and without prior use of oxygen, associated with three common chronic respiratory diseases: Chronic Obstructive Pulmonary Disease (COPD, *n* = 24/20), obstructive sleep apnea (OSA, *n* = 41/39) and interstitial lung diseases (ILD, *n* = 5/2), and 70 healthy subjects covering among all, a broad spectrum of SaO_2_ and PaO_2_. The patients were selected from the corresponding INER clinics: sleep disorders clinic; pulmonary fibrosis clinic; and COPD and smoking cessation clinic; and had not had a recent (1 month) exacerbation of the disease, infection, or hospitalization or used supplemental oxygen.

The FEV_1_/FVC ratio is a standard criterion for airflow obstruction. In the presence of airflow obstruction, the volume exhaled in the first second (FEV_1_) decreases in proportion to the total volume exhaled (FVC). The COPD patients all had FEV_1_/FVC ratio <0.7. and had chronic exposure to smoking or biomass-burning smoke while cooking. Those with interstitial diseases had a diagnosis of idiopathic pulmonary fibrosis or fibrosis associated with rheumatic diseases. Obstructive sleep apnea patients had been evaluated with respiratory polygraphy with a respiratory index of >15 or polysomnography with an apnea-hypopnea index >15.

The healthy controls were generally active or retired INER workers and their relatives; had a normal oxygen saturation by pulse oximetry (>92%), lacked a history of respiratory diseases, and acute and chronic respiratory symptoms and signs during the last 15 days; and they had never smoked.

Both patients and healthy participants were residents of the Valley of Mexico, at an average altitude of 2240 m above sea level for at least 5 years. The average barometric pressure is 585 mm Hg, and the partial pressure of oxygen in the inspired air, humid, and at body temperature, is about 113 mm Hg.

After signing the informed consent, general health, medical history, and symptoms questionnaire were applied at the first visit, including questions to determine the clinical characteristics of hypoxemia and respiratory risk factors. Baseline measurements were taken that included weight, height, body mass index (weight/ height^2^ in kg/meters^2^), and pulse oximetry at rest (five measurements and the average is reported). Spirometry was also performed, in a sitting position before and 15 min after a bronchodilator (400 mcg of salbutamol with a metered-dose aerosol and large volume spacer) [11]), to obtain three acceptable maneuvers with the two best FEV1 and FVC within 150 mL, following the American Thoracic Society/European Thoracic Society (ATS/ERS) procedures [12].

Arterial blood samples were taken at rest, from the radial artery in the non-dominant hand with a 25–27 gauge needle, and processed immediately on an ABL800 FLEX analyzer (Radiometer, Copenhagen, Denmark). Healthy volunteers had only the pulse oximetry without an arterial blood sample. PaO_2_ was classified as normal at the recruitment time if 60+ mm Hg, borderline if 55–59 mm Hg, and hypoxemia if <55 mm Hg.

### 2.1. Serum Protein Determination

The determination in the serum of the following proteins was carried out: hypoxia-inducible factor 1-alpha (HIF-1α), vascular endothelial growth factor (VEGF), and erythropoietin (EPO) using the commercial kits: HIF1A (Cat. number EHIF1A2. Invitrogen, Bender MedSystems GmbH. Vienna, Austria), human VEGF (Cat. number KHG0111. Novex^®^ Life Technologies, Waltham, MA, USA), and human Erythropoietin ELISA (Cat. number BMS2035. Invitrogen, Waltham, MA, USA). The manufacturer’s instructions for processing were followed. The kit detection levels for HIF-1α are 82–20,000 pg/mL, with a sensitivity ≤30 pg/mL, for VEGF 23.4–1500 pg/mL, with a sensitivity <5 pg/mL, EPO of 1.8–100 mIU/mL, and 0.17 mIU/mL sensitivity.

### 2.2. Ancestry Informative Markers (AIMs) Genotyping

Allelic discrimination of 14 single nucleotide polymorphisms (SNPs) was carried out through qPCR, using TaqMan probe assays: rs4528122, rs986690, rs10516422, rs10515716, rs1878071, rs4084051, rs7853112, rs10511491, rs1039336, rs1479514, rs147756, rs14780714, and rs147756. The DNA was adjusted to 15 ng/µL; a reaction mixture was prepared with TaqMan probe and Master Mix TaqMan™ (Applied Biosystems, Waltham, MA, USA), as well as nuclease-free water. It was mixed and centrifuged at 1500 rpm to run in a 7300 Real-Time PCR system thermal cycler.

Patients meeting criteria for hypoxemia for oxygen prescription were loaned an oxygen concentrator to use continuous oxygen at home. According to the treating physician’s prescription, sleep apnea patients similarly received an automatic-CPAP device for treatment (with or without additional oxygen).

After a month, the patients returned to retake a venous blood sample and baseline studies.

### 2.3. Statistical Analysis

The degree of hypoxemia quantified through PaO_2_ was correlated with the variables of response to hypoxemia, including hemoglobin levels, hematocrit, erythropoietin, and HIF-1α.

A comparison analysis was performed within groups (before and after treatment with supplemental O2) and between groups of patients and controls, using the Mann-Whitney and Kruskal-Wallis U statistical tests after Bonferroni correction. Spearman’s correlation analysis was performed using the patients’ clinical and demographic variables and the proteins determined at baseline and follow-up. They were carried out in the RStudio v3.5.2 environment. On the other hand, the principal component analysis (PCA) was performed, taking the eigenvalues 1 and 2 of the 14 genotyped SNPs, using a 95% quality control, in the PLINK v1.07 and EIGENSTRAT v3.0 software. The rest of the analyses were carried out using STATA statistical software [13].

## 3. Results

At the basal time, seventy patients with respiratory diseases were recruited (63% men, with age 59.7 ± 11.7 and PaO_2_ of 53.4 ± 4.6, SaO_2_ of 86.3 ± 3.3) and 70 healthy controls (62% men and age of 56 ± 12.2 years and SpO_2_ 94.2 ±1.9). At the follow-up, 20 patients with COPD (age 68.4) were recruited, 39 with obstructive sleep apnea (age 53), and 2 with interstitial lung diseases (age 63.8). Table 1 shows clinical data for each group.

### 3.1. Protein Measurements

The serum protein levels obtained for the three proteins in the baseline determination (without supplemental oxygen treatment) shown HIF-1α and VEGF were increased in the control group compared with the patients’ groups; however, differences were not statistically significant (*p*-value = 0.509 and 0.091, respectively).

EPO levels were higher in patients than in the control group, where a statistically significant difference was found in the OSA *vs*. the control group (16.80 vs. 5.98, *p* < 0.001 after Bonferroni correction). The protein levels for each study group are shown in Figure 1.

The patients’ intragroup analyses before and after treatment are shown in Table 2; HIF-1α at follow-up decreased in patients with COPD and OSA and increased in patients with fibrosis. In the VEGF levels, no differences were found between the groups before and after. Finally, EPO levels decreased at follow-up both in the group in general and in patients with COPD and OSA (*p* < 0.05).

The graphs in Table 2 show the change in each group’s levels at baseline and follow-up and each protein’s levels in the patient groups.

#### 3.1.1. Correlation Analysis

A correlation analysis was carried out between the protein levels (before and after the supplemental oxygen treatment) and clinical variables, as hematic cytometry and blood chemistry. Figure 2, correlations: baseline EPO with age (r^2^ = −0.38) and eosinophil values (r^2^ = 0.3); on the other hand, EPO at follow-up is negatively correlated with oxygen saturation (r^2^ = −0.45) and positive with the smoking index (r^2^ = 0.31). And moderate positive and negative correlations (<0.6) with hemoglobin concentration and VEGF and EPO levels both at baseline and at follow-up of the patients, as well as low correlations (<0.4) with blood chemistry analytes such as uric acid, glucose, and creatinine.

#### 3.1.2. Ancestral Contribution According to AIMs

Appendix A Appendix A shows the SNPs selected according to the presence in the native and Mexican mestizo populations; the SNPs with a delta value > 0.4 are taken for the ancestry analysis. The principal component analysis was performed, taking as reference populations Caucasian European residents of Utah (CEU) and Amerindian Zapotec from Oaxaca (ZAP). The principal components graph was obtained, which shows the distribution of the 92 individuals AIMs’ in the study groups, which gave rise to Mexican mestizos; it is observed that the distribution between healthy individuals and cases is similarly displaced along the vectors.

## 4. Discussion

In the longitudinal results shown, before and after using oxygen for 1 month, hypoxemic patients did not have on average increased levels of HIF-1α, but after oxygen therapy, the HIF-1α decreased in COPD and OSA but not in fibrosis, which may be related to the course of deterioration that is common in pulmonary fibrosis or to the pathogenesis of the disease. EPO was raised in patients and decreased with oxygen and could be a useful marker of oxygenation. VEGF leading to vascular proliferation and remodeling had serum levels heterogeneous in the sample studied. The control group and COPD patients have similar levels, slightly increased in the OSA group but decreased in the ILD group, and remained similar after oxygen supplementation. This pattern could be a mixture of the direct impact of hypoxemia, treatment with oxygen, and different pathogenesis of the diseases. The possible impact of infection or inflammatory diseases was reduced as much as possible by checking on comorbidities and avoiding exacerbations.

HIF-1α levels were reported to increase in COPD patients in an amount proportional to severity [14] and increased in sleep apnea with hypoxemia [15]. Many oxygen-sensitive regulatory mechanisms work through HIF-1α, and recent literature regarding the hypoxic stimulus and its pathological implications deals primarily with HIF-related findings. HIF-1α is pivotal in the adaptation to chronic hypoxia: it induces gene expression for fructose-2-6-biphosphatase, an enzyme-switching glucose metabolism towards glycolysis, allowing energy production in anaerobic conditions [16].

Serum erythropoietin was previously found increased in COPD patients with severe nocturnal and diurnal hypoxemia [17], and patients with severe COPD [18], although the response to EPO in COPD appears to be blunted [19]. EPO, in general, drops with hypoxemia treatment in COPD [20] and sleep apnea [21]. Erythropoietin is an endogenous glycoprotein hormone primarily produced by the kidneys in response to a decrease in tissue oxygenation tension. EPO acts as the primary stimulus for erythropoiesis in normal conditions, promoting the differentiation of progenitor cells into erythrocytes. Several studies have demonstrated that COPD anemic patients frequently present with a significant enhancement in erythropoietin levels [22]. VEGF was increased in hypoxemic patients with COPD during an exacerbation [23].

In populational studies, it has been shown that the primary determinant of the oxygenation status is the altitude above sea level, so that the higher the altitude, the more oxygen therapy required [10]. In Mexico City (2240 m of altitude), this has been quantified, and up to 6% of the population over 40 years of age would have SO indication if we used the criteria generated at sea level [10]. However, the current criteria to establish the hypoxemia diagnosis for the SO prescription was generated 30 years ago and originated from studies carried out at sea level in COPD patients [1,2]. These criteria, based on PaO_2_, prevail worldwide without considering the disease or altitude of residence. Altitude could generate a better adaptation to hypoxemia, especially over many generations, and prepare inhabitants to tolerate lower levels of PaO_2_ than if hypoxemia happens to residents of sea level. Indeed, a fraction of that 6% of residents of Mexico City, at least 40 years of age, with SpO_2_ ≤ 88%, require SO, but among the group could be individuals who are adapted to living at altitude and in whom the benefits of administering SO would be minimal and at the expense of the high cost of SO therapy. Having additional markers of hypoxemia (other than PaO_2_, polycythemia, and pulmonary hypertension) may provide clinical tools to better decide if a resident at moderate altitude is presenting a bodily response to hypoxemia and is at risk of complications, and in the best scenario, the marker pattern predicts adverse outcomes that would be prevented with supplementary oxygen.

In summary, in this pilot study, EPO promises to help identify hypoxemia early and respond to oxygen treatment, except in pulmonary fibrosis that tends to worsen. HIF-1α would likely be more useful in patients’ follow-up, although again in fibrosis tends to increase despite oxygen treatment.

VEGF is decreased in patients with fibrosis, increases with time with oxygen therapy, and seems more linked to the course of the disease than to hypoxemia.

## 5. Conclusions

Erythropoietin and HIF-1α identified hypoxemia initially and responded to oxygen therapy. In pulmonary fibrosis, HIF-1α, EPO, and VEGF increased with oxygen therapy, which is likely more linked to the disease’s pathogenesis and course than to hypoxemia.

## Figures and Tables

**Figure 1 life-11-00428-f001:**
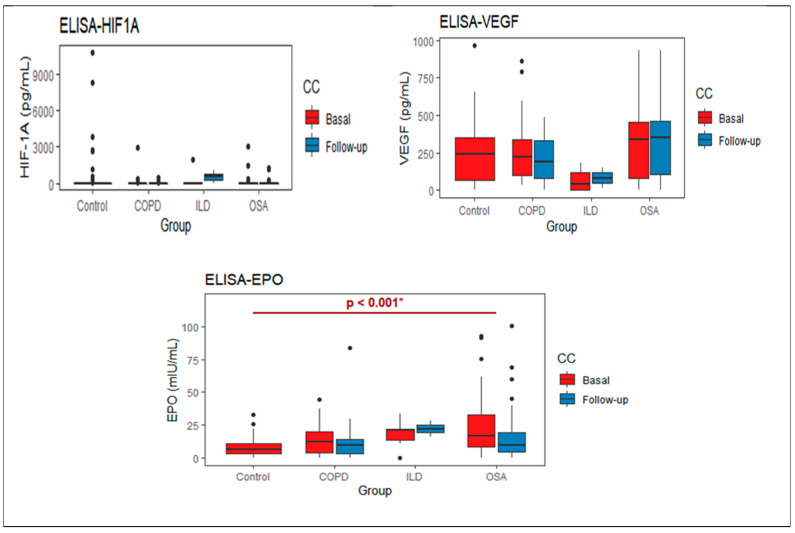
Basal and follow-up protein levels for each study group. * *p*-value obtained by the Kruskall Wallis test, poshoc with the Bonferroni test, with difference between Control- OSA.

**Figure 2 life-11-00428-f002:**
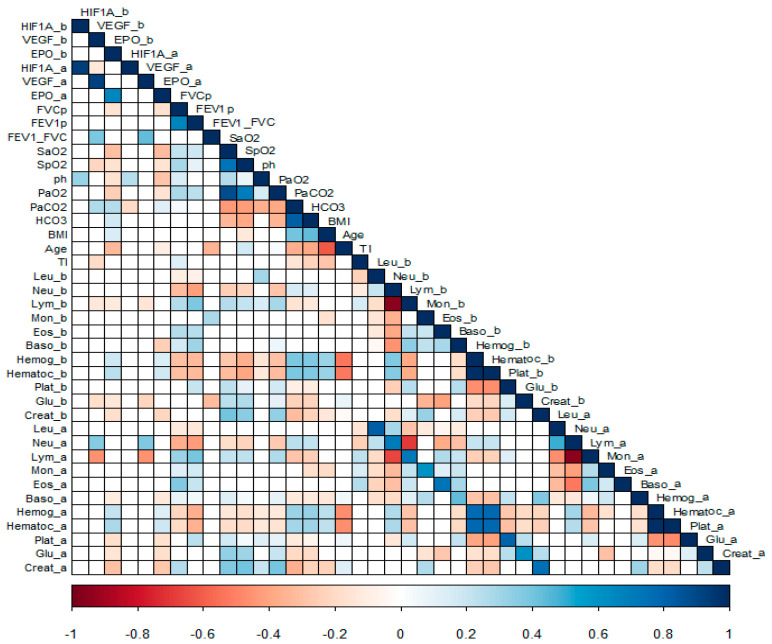
Correlations of proteins determined HIF-1α, VEGF, and EPO in patients before and after treatment with O2 with clinical variables of interest. The correlations with *p*-value < 0.05 are shown in color (intensities from red to blue), the boxes in white have a value of *p* > 0.05. _b: before, _a: after, TI: Tobacco index, Leu: leukocytes, Neu: neutrophils, Lym: Lymphocytes, Mon: Monocytes, Eos: Eosinophils, Bas: Basophils, Hemog: Hemoglobin, Hematoc: Haematocrit, Plat: Platelets, Glu: Glucose, Creatinine; the plot shows only significant levels of correlation in color red to blue.

**Table 1 life-11-00428-t001:** Clinical and demographical data for each group in the follow-up.

	COPD	OSA	ILD	Controls
**Sex (M|F)**	12|8	27|12	1|1	44|26
**Age (years)**	68 (63–71)	53 (42–57)	64 (62–63)	56 (45–69)
**BMI (kg/m^2^)**	24.50 (22.23–27.25)	39 (36.50–44.50)	24 (23–28)	26 (24–28.15)
**Pulse** **Oximetry (%)**	89 (87–91)	88 (87–90)	94 (93-94)	94 (93.25–95.90)
**PaO_2_ (mmHg)**	54.95 (51.25–56.40)	53.95 (49.67–56.77)	47.90 (47.70–55.60)	NT
**SpO_2_ (%)**	88 (86–88)	87 (77–88)	85 (77–87)	NT
**SaO_2_ (%)**	87.47 (84.47–88.78)	86.90 (84.79–88.35)	83.90 (82.10–85.00)	NT
**FEV_1_ (%)**	48 (46.50–64.50)	81.5 (72.75–94.0)	61 (39.5–83.5)	108 (98–117)
**FVC (%)**	79 (72.50–96.50)	88 (76–94)	54.50 (37.00–72.75)	112.50 (97.25–119.75)
**DLCO (%)**	82.50 (64.75–104.50)	123 (103–139.8)	45 (39–46)	113.5 (105.8–127)

The data of the quantitative variables are shown in median (interquartile range) and categorical variables in n. PaO_2_: Arterial oxygen pressure; SpO_2_: Oxygen saturation; FVC: Forced vital capacity; FEV1: Forced expiratory volume in the first second. Not tested.

**Table 2 life-11-00428-t002:** Intragroup analysis, before and after oxygen treatment.

	Basal	Follow Up	*p*-Value	
	HIF1A		0.270	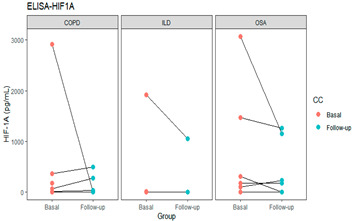
COPD	146.5 (0–2917.1)	39.62 (0–491.03)	0.324	
OSA	125.0 (0–3071.8))	71.96 (0–1260.26)	0.389	
ILD	384.87 (0–1917.95)	526.3 (263.1–789.4)	0.665	
	VEGF		0.677	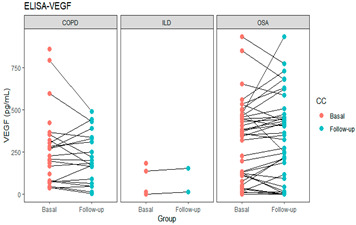
COPD	216.07 (78.83–328.81)	192.47 (83.36–331.61)	0.331	
OSA	342.0 (78–457.20)	350.4 (104.1–464.1)	0.678	
ILD	12.6 (1.1–137.46)	83.36 (48.13–118.59)	0.809	
	EPO		0.033	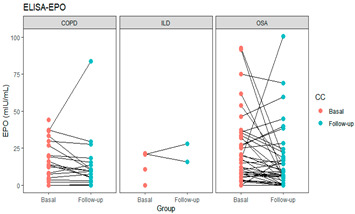
COPD	12.39 (4.19–21.98)	9.44 (2.93–14.19)	0.023	
OSA	16.80 (7.96–32.56)	9.20 (4.04–19.01)	0.025	
ILD	20.72 (10.62–21.04)	21.98 (18.98–24.98)	0.809	

Protein levels in median (Interquartile range), the upper *p*-value shows the comparison of before and after treatment (independent of diseases); additionally, the *p*-value obtained before and after each disease.

## Data Availability

The data presented in this study are available in Appendix A Appendix A.

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
