# Peer review of "Clinical Markers of Chronic Hypoxemia in Respiratory Patients Residing at Moderate Altitude"

_life, 2021, doi:10.3390/life11050428_

Round 1

Reviewer 1 Report

In a manuscript by Fernandez-Plata et al, the authors aimed to identify markers of hypoxemia that could be used in the evaluation of pts addition to pulse oximetry and arterial blood gases, especially those that could respond after 1 month of oxygen use. In their pilot study, 24 had COPD, 41 had OSA, and 5 with ILD were enrolled, all of them having chronic hypoxemia, as well as 70 healthy subjects as controls. Markers evaluated included HIF-1α, VEGF and  EPO. They concluded that erythropoietin and HIF-1α identified hypoxemia initially and responded to oxygen. In pulmonary fibrosis HIF-1α, EPO and VEGF increased with oxygen therapy, likely linked to the disease's pathogenesis and clinical course rather than to hypoxemia.

My comments:

  1. Please check again the number of pts enrolled (45vs 46 and 45 v 70 controls). These data are inconsistent throughout the abstract methods and results section.
  2. Therefore, I suppose, the data should be reassessed once again.
  3. Discussion section is way to short.

Author Response

Thank you very much for your kind reviewer report; this allowed us to improve redaction and correct some mistakes.

Regarding the number of enrolled participants, we have reviewed and corrected it across the manuscript; now, you can find it in red text to easy location.

Also, the discussion section was improved according to the main results.

Reviewer 2 Report

The authors aimed to identify markers of hypoxemia that can be used in the evaluation of patients after oxygen therapy. The aim is simple and clear. Even though the subject is interesting and may be clinically relevant since it may lead to suitable treatment of hypoxemic patients, the study does not provide any sufficient data to conclude.

MAJOR COMMENTS:

1) ABSTRACT. The title announces to speak about patients living at moderate altitude but there is no mention of altitude in the abstract. However, patients residing at altitude do not have the same clinical profile, especially the basal concentration of proteins involved in the hypoxic response. Plus, the authors explain in the discussion that altitude must be considered in the establishment of the hypoxemia diagnosis (criteria generated at sea level).

2) INTRODUCTION. The introduction does not provide sufficient background about the hypobaric hypoxia found at altitude and the resulting modification of oxygen partial pressure.

It never mentions that Mexico City is at altitude 2240m, which is equivalent to an atmospheric pressure of 600 mmHg (=0.8 bar) and an oxygen partial pressure of 116 mmHg (149 mmHg at sea level, i.e. a decrease of 22%).

There are no relevant references about HIF-1alpha (i.e. G. Semenza).

The introduction does not refer to criteria used to establish the hypoxemia diagnosis, so there is no real link between introduction and discussion.

3) METHODS. FEV1/FVC ratio and respiratory index must be briefly explained.

4) RESULTS. The total number of patients and the distribution between groups are a bit confusing (abstract vs methods vs results). How many patients are there in total? How many controls? It is necessary to harmonize.

It is very surprising to note that figure 2 is not in English. I had to look up the translation of some words. The authors have to present figures in English. Besides, the text font is too small.

Table 1. shows the baseline protein levels comparison between groups

Figure 1. shows the basal and follow-up protein levels for groups

Table 2. shows same results as table 1. but without controls

Figure 2. shows correlations of proteins with clinical variables. Maybe the authors could emphasize the EPO result as it is the most relevant.

I think it is essential to reorganize the results. The authors speak about Table 2 before Table 1. Maybe the authors would reconsider the number of figures? It seems a little redundant.

I think the supplementary table 1. is unnecessary. The authors should give the values in the text.

5) DISCUSSION/CONCLUSION

The authors rightly discuss the consideration of altitude in the establishment of the hypoxemia diagnosis (criteria generated at sea level).

I think this part of the reflection is the most important in this article and the author should consider to modify the manuscript in that perspective.

Author Response

Thank you very much for your support in the review of our manuscript; now we have addressed all your commentaries and suggestions.

1) ABSTRACT. Now is included, please find highlighted in red text.

2) INTRODUCTION. Thank you for this excellent observation; now you can see that we have added some lines regarding the altitude, criteria used to establish the hypoxemia diagnosis, and one else Semenza [6] reference about HIF-1alpha.

3) METHODS. Now, we have included a brief description of the FEV1/FVC ratio.

4) Thank you for your observation, now is corrected in the different manuscript's sections.

Figure 2 now is in English and modified for better visualization. Also was summarized in a single figure, attending your suggestion to increases the size text font and reorganization.

The supplementary table 1 was deleted and information included in the main text.

5) DISCUSSION/CONCLUSION. Now, this section was modified according to your suggestion. Thank you very much!

Round 2

Reviewer 1 Report

Thank you. I have no further remarks.

Author Response

Thank you for your kind review, we really appreciate your time and effort to collaborate in the editorial process.

Reviewer 2 Report

General comment : The manuscript has been improved, the requested changes have been made, especially in the introduction and the discussion which are really imporved. Thank you. However, there are still some clarifications to be made.

1) Methods: about the distribution of patients between groups, maybe you can clearly indicate the number of patients having COPD before and after the oxygenation. It should be written out in full at least at the first time because it is not clear enough...

2) Results : Have you considered summarising the characteristics of patients in a table? age, sex, PaO2, SpO2, FEV etc.

I think it could be more clear if you insert a single table with the protein measurement before and after (baseline vs follow up)

In summary, I think authors should give in Table 1 the characteristics of patients and in Table 2 all the protein measurements results.

The authors speak about Table 2 before Table 1 (page 4, line 172 do you mean Table 1??)

Author Response

General comment : The manuscript has been improved, the requested changes have been made, especially in the introduction and the discussion which are really imporved. Thank you. However, there are still some clarifications to be made.

A: Thank you for your comment; we appreciate your time and effort in reviewing our manuscript.

1) Methods: about the distribution of patients between groups, maybe you can clearly indicate the number of patients having COPD before and after the oxygenation. It should be written out in full at least at the first time because it is not clear enough...

A: Thank you, now we have reformulated.

2) Results : Have you considered summarising the characteristics of patients in a table? age, sex, PaO2, SpO2, FEV etc.

I think it could be more clear if you insert a single table with the protein measurement before and after (baseline vs follow up).

A: After attending your kind suggestions, this information can be found in Table 3.

In summary, I think authors should give in Table 1 the characteristics of patients and in Table 2 all the protein measurements results.

A: We particularly thank this comment, which allows us to show clinical and demographical data for included groups in an independent mode.

The authors speak about Table 2 before Table 1 (page 4, line 172 do you mean Table 1??)

A: Attending your comment, we deleted this Table and briefly mentioned this data in the text, which also can be seen in Table 3.

This manuscript is a resubmission of an earlier submission. The following is a list of the peer review reports and author responses from that submission.